# High-resolution mapping reveals hundreds of genetic incompatibilities in hybridizing fish species

Molly Schumer[1]*, Rongfeng Cui[2,3], Daniel L Powell[2,3], Rebecca Dresner[1], Gil G Rosenthal[2,3], Peter Andolfatto[1,4]

[1]Department of Ecology and Evolutionary Biology, Princeton University, Princeton, United States; [2]Department of Biology, Texas A&M University, College Station, United States; [3]Centro de Investigaciones Científicas de las Huastecas 'Aguazarca', Calnali, Mexico; [4]Lewis-Sigler Institute for Integrative Genomics, Princeton University, Princeton, United States

**Abstract** Hybridization is increasingly being recognized as a common process in both animal and plant species. Negative epistatic interactions between genes from different parental genomes decrease the fitness of hybrids and can limit gene flow between species. However, little is known about the number and genome-wide distribution of genetic incompatibilities separating species. To detect interacting genes, we perform a high-resolution genome scan for linkage disequilibrium between unlinked genomic regions in naturally occurring hybrid populations of swordtail fish. We estimate that hundreds of pairs of genomic regions contribute to reproductive isolation between these species, despite them being recently diverged. Many of these incompatibilities are likely the result of natural or sexual selection on hybrids, since intrinsic isolation is known to be weak. Patterns of genomic divergence at these regions imply that genetic incompatibilities play a significant role in limiting gene flow even in young species.

*For correspondence: schumer@princeton.edu

Competing interests: The authors declare that no competing interests exist.

## Introduction

Hybridization between closely related species is remarkably common (*Mallet, 2005*). Many hybridizing populations and species remain genetically and ecologically distinct despite bouts of past admixture (e.g., *Scascitelli et al., 2010*; *Vonholdt et al., 2010*). This has led to a surge of interest in identifying which and how many loci are important in maintaining species barriers. Recent work has focused on identifying so-called 'genomic islands' of high divergence between closely related species (e.g., *Turner et al., 2005*; *Nadeau et al., 2012*). This approach assumes that the most diverged regions between species are most likely to be under divergent selection between species or important in reproductive isolation. However, divergence-based measures need to be interpreted with caution because they are susceptible to artifacts as a result of linked selection events (including background selection and hitch-hiking) such that outlier regions might reflect low within-population polymorphism rather than unusually high divergence (discussed in *Charlesworth, 1998*; *Noor and Bennett, 2009*; *Renaut et al., 2013*), and there are many possible causes of elevated divergence that are not linked to isolation between species.

Investigating genome-wide patterns in naturally occurring or laboratory-generated hybrid populations is another approach to characterize the genetic architecture of reproductive isolation (*Payseur, 2010*). Hybridization leads to recombination between parental genomes that can uncover genetic incompatibilities between interacting genes. When genomes diverge in allopatry, substitutions that accumulate along a lineage can lead to reduced fitness when hybridization decouples them from the genomic background on which they arose. The best understood of these epistatic interactions, called

**eLife digest** In nature, closely related species often interbreed to produce hybrids. However, hybrids are often less fertile or unable to compete with parent species, making them less likely to thrive in the wild.

When the genomes of two different species are mixed, versions of genes that are meant to work together can become separated, which means that these genes do not work as well as they should. This reduces the hybrids' chances of survival, and a poor survival rate of hybrids is one barrier that keeps different species distinct, even though the species can interbreed.

Two species of swordtail fish live in the rivers in Mexico, and although they mostly live in different stretches of these rivers, the two species interbreed to produce hybrids in the regions where they overlap. These hybrids can outnumber the parental species in these 'hybrid zones', but the two species have remained separate in other parts of the rivers. Though some genetic incompatibilities that might keep the species distinct have previously been suggested, it is not known how many incompatibilities there are in these fish's genomes.

Schumer et al. have searched the genomes of wild hybrids between these species and found hundreds of genetic incompatibilities. These were identified by looking for species-specific pairs of genes that are found together more often than would be expected if there were no selection against hybrids. It is likely that many of these incompatibilities reduce the evolutionary fitness of the hybrid fish and Schumer et al. suggest that many could be the result of environmental pressures and the fish's mating preferences. Furthermore, Schumer et al. demonstrate that genes close to identified incompatibilities are more different between species, on average, than genes that are further away. When there is on-going interbreeding between two the species (as is the case with the swordtails), this finding is expected only if these incompatibilities reduce the hybrids' chances of finding mates or surviving.

The findings of Schumer et al. demonstrate how conflicts in the genomes of two species allow these species to remain distinct even when they live in overlapping environments and frequently interbreed. Future work will investigate how these genetic incompatibilities shape the hybrid populations that are found in the wild; and which incompatibilities are caused by poor survival of the hybrids or by the fish's mating preferences selecting against the hybrids.

'Bateson-Dobzhansky-Muller' (BDM) incompatibilities (*Coyne and Orr, 2004*), can occur as a result of neutral substitution or adaptive evolution, and are thought to be common based on theoretical (*Orr, 1995*; *Turelli et al., 2001*) and empirical studies (*Presgraves, 2003*; *Presgraves et al., 2003*; *Payseur and Hoekstra, 2005*; *Brideau et al., 2006*; *Sweigart et al., 2006*). One defining feature of BDM incompatibilities is that they are predicted to have asymmetric fitness effects in different parental backgrounds, such that only a subset of hybrid genotypes are under selection. Though the BDMI model is an important mechanism of selection against hybrids, other evolutionary mechanisms can contribute to hybrid incompatibility. For example, co-evolution between genes can result in selection on all hybrid genotype combinations due to the accumulation of multiple substitutions (*Seehausen et al., 2014*). Similarly, natural or sexual selection against hybrid phenotypes can be considered a form of hybrid incompatibility; in this case the genotypes under selection will depend on their phenotypic effects.

How common are hybrid incompatibilities and what is their genomic distribution? Most studies to date have addressed this question by mapping hybrid incompatibilities that contribute to inviability or sterility (*Orr, 1989*; *Orr and Coyne, 1989*; *Barbash et al., 2003*; *Presgraves, 2003*; *Presgraves et al., 2003*), in part because these incompatibilities affect hybrids even in a lab environment. Initial genome-wide studies in *Drosophila* and other organisms suggest that the number of incompatibilities contributing to hybrid viability and sterility can range from a handful to hundreds accumulating between deeply diverged species (*Harushima et al., 2001*; *Presgraves, 2003*; *Moyle and Graham, 2006*; *Masly and Presgraves, 2007*; *Ross et al., 2011*); recent work has also suggested that substantial numbers of incompatibilities segregate within species (*Cutter, 2012*; *Corbett-Detig et al., 2013*). However, because research has focused primarily on postzygotic isolation, little is known about the total number of loci contributing to reproductive isolation between species. For example, research shows that selection against hybrid genotypes can be strong even in the absence of postzygotic

isolation (*Fang et al., 2012*). Thus, the focus on hybrid sterility and inviability is likely to substantially underestimate the true number of genetic incompatibilities distinguishing species.

If negative epistatic interactions are important in maintaining reproductive isolation, specific patterns of genetic variation are predicted in hybrid genomes. In particular, selection against hybrid individuals that harbor unfavorable allele combinations in their genomes will lead to under-representation of these allelic combinations in a hybrid population. Thus, selection can generate non-random associations, or linkage disequilibrium (LD), among unlinked loci in hybrid genomes (*Karlin, 1975*; *Hastings, 1981*). Patterns of LD in hybrid populations can therefore be used to identify genomic regions that are important in establishing and maintaining reproductive isolation between species.

Only a handful of studies have investigated genome-wide patterns of LD in hybrid populations. *Gardener et al. (2000)* evaluated patterns of LD at 85 widely dispersed markers (~0.03 markers/Mb) in sunflowers and found significant associations among markers known to be related to infertility in hybrids. Similarly, *Payseur and Hoekstra (2005)* evaluated patterns of LD among 332 unlinked SNPs (~0.12 markers/Mb) in inbred lines of hybrid mice and identified a set of candidate loci with strong conspecific associations. More recently, *Hohenlohe et al. (2012)* investigated genome-wide patterns of LD at ~2000 sites (~4.5 markers/Mb) in oceanic and freshwater sticklebacks and found two unlinked regions in strong LD that are highly differentiated between populations. These studies suggest hybridization can expose the genome to strong selection that leaves detectable signatures of LD in hybrids.

In this study, we evaluate genome-wide patterns of LD in replicate hybrid populations of two species of swordtail fish, *Xiphophorus birchmanni* and *X. malinche*. These species are recently diverged (0.5% genomic divergence per site; 0.4% genomic divergence following polymorphism masking) and form multiple independent hybrid zones in river systems in the Sierra Madre Oriental of Mexico (*Culumber et al., 2011*). *X. malinche* is found at high elevations while *X. birchmanni* is common at low elevations; hybrids occur where the ranges of these two species overlap. The strength of selection on hybrids between these two species is unknown, but several lines of evidence have suggested that selection may be weak. Hybrids are abundant in hybrid zones, often greatly outnumbering parental individuals. Hybrids are tolerant of the thermal environments at the elevations in which they are found (*Culumber et al., 2012*). Though there is some evidence of BDMIs between the species that cause lethal melanomas, these melanomas typically affect hybrids post-reproduction (*Schartl, 2008*) and may constitute a weak or even favorable selective force (*Fernandez and Morris, 2008*; *Fernandez and Bowser, 2010*). Finally, recent behavioral studies show that once hybrids are formed, hybrid males actually have an advantage due to sexual selection compared to parental individuals (*Figure 1*; and see *Fisher et al., 2009*; *Culumber et al., in press*). However, the genomes of adult hybrids sampled from hybrid populations have already been subject to multiple generations of selection (at least 30; *Rosenthal et al., 2003*), making it difficult to evaluate the extent of selection on hybrids without genetic information. By surveying the genomes of hybrids from natural populations, we are able to identify interacting genomic regions under selection in hybrids, giving a powerful picture of the number of regions involved in reproductive isolation between this recently diverged species pair.

To evaluate genome-wide patterns of LD in hybrid populations, we further develop the multiplexed shotgun genotyping (MSG) protocol of *Andolfatto et al. (2011)*. Originally developed for QTL mapping in controlled genetic crosses, we describe modifications that make the technique applicable to population genetic samples from hybrid populations. With this approach, we assign ancestry to nearly 500,000 ancestry informative markers throughout the genome, allowing us to evaluate genome wide patterns of LD at unprecedented resolution (~820 markers/Mb). The joint analysis of two independent hybrid zones allows us to distinguish the effects of selection against genetic incompatibilities from confounding effects due to population history (*Gardner et al., 2000*). We further evaluate loci that are in significant LD to investigate the mechanisms of selection on these pairs. Our results support the conclusion that a large number of loci contribute to reproductive isolation between these two species, that these loci have higher divergence than the genomic background, and that selection against genetic incompatibilities maintains associations between loci derived from the same parental genome.

## Results

### Hybrid zones have distinct demographic histories

We used a modified version of the MSG analysis pipeline, optimized for genotyping in natural hybrids ('Materials and methods', *Figure 1—figure supplement 1*) to genotype individual fish collected from two

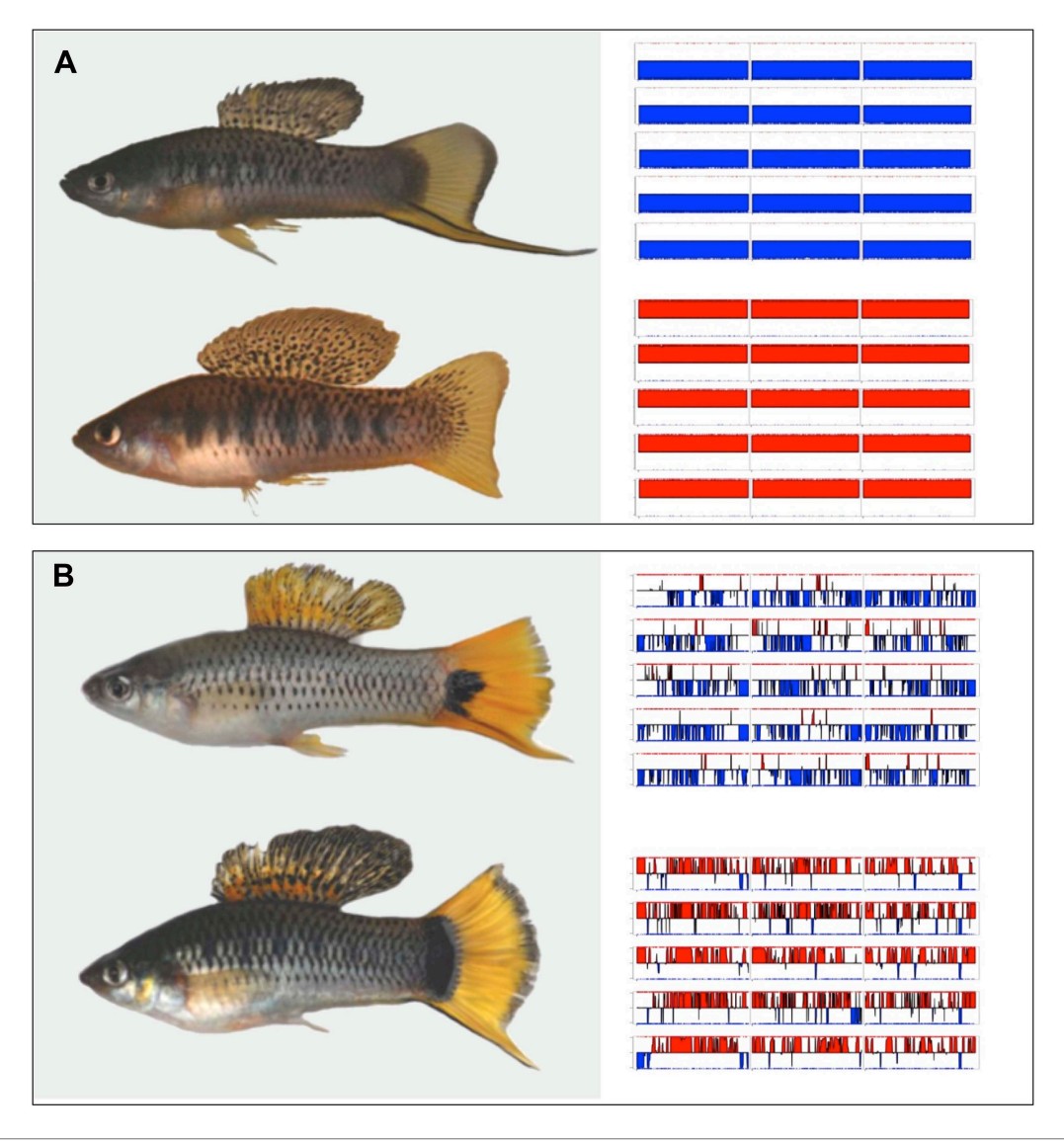

**Figure 1**. Hybrids between *X. malinche* and *X. birchmanni*. (**A**) Parental (*X. malinche* top, *X. birchmanni* bottom) and (**B**) hybrid phenotypes with sample MSG genotype plots for linkage groups 1–3 (see ***Figure 1—figure supplement 1*** for more examples) for each population shown in the right panel. In MSG plots, solid blue indicates homozygous *malinche*, solid red indicates homozygous *birchmanni* and regions of no shading indicate heterozygosity. Hybrid individuals from Tlatemaco (**B** top) have *malinche*-biased ancestry (solid blue regions) while hybrids from Calnali (**B** bottom) have *birchmanni*-biased ancestry (solid red regions).

The following figure supplements are available for figure 1:

**Figure supplement 1**. MSG ancestry plots for parental and hybrid individuals.

independently formed hybrid zones, Calnali and Tlatemaco (***Figure 1***, ***Culumber et al., 2011***). We genotyped 143 hybrid individuals from Calnali, 170 hybrids from Tlatemaco, and 60 parents of each species, determining ancestry at 469,400 markers distinguishing *X. birchmanni* and *X. malinche* at a median density of 1 marker per 234 bp ('Materials and methods'). On average, hybrids from the Calnali hybrid zone derived only 20% of their genome from *X. malinche*, while hybrids from the Tlatemaco hybrid zone had 72% of their genomes originating from *X. malinche*. Most hybrid individuals were close to the average hybrid index in each group (Tlatemaco: 95% of individuals 66–80% *malinche* ancestry; Calnali: 95% of individuals 14–35% *malinche* ancestry). We determined the time since hybridization based on the decay in

linkage disequilibrium ('Materials and methods'), assuming an average genome-wide recombination rate of 1 cM/378 kb (*Walter et al., 2004*). Estimates of hybrid zone age were similar for the two hybrid zones (Tlatemaco 56 generations, CI: 55–57, Calnali 35 generations CI: 34.5–35.6). Interestingly, these estimates are remarkably consistent with available historical estimates, which suggest that hybridization began within the last ~40 generations due to disruption of chemical cues by pollution (*Fisher et al., 2006*).

## Significant LD between unlinked genomic regions

Ancestry calls at 469,400 markers genome-wide were thinned to 12,229 markers that are sufficient to describe all changes in ancestry (±10%) across individuals in both populations ('Materials and methods'). Using this thinned data set, we analyzed patterns of LD among all pairs of sites. The physical distance over which $R^2$ decayed to <0.5 was approximately 300 kb in both populations (*Figure 2—figure supplement 1*). Average genome-wide $R^2$ between physically unlinked loci was 0.003 in Tlatemaco and 0.006 in Calnali (*Figure 2A*), and did not significantly differ from null expectations (1/2n, where n is the sample size). A p-value for the $R^2$ value for each pair of effectively unlinked sites was estimated using a Bayesian Ordered Logistic Regression t-test (*Figure 2B*, 'Materials and methods').

The expected false discovery rates (FDRs) associated with given p-value thresholds applied to both populations were determined by simulation ('Materials and methods', *Figure 3*, *Figure 3—figure supplement 1*). Using data from two hybrid populations to examine LD circumvents two problems: (1) within a single population cases of LD between unlinked sites are unlikely to be strong enough to survive false discovery corrections and (2) even interactions that are highly significant could be caused by population

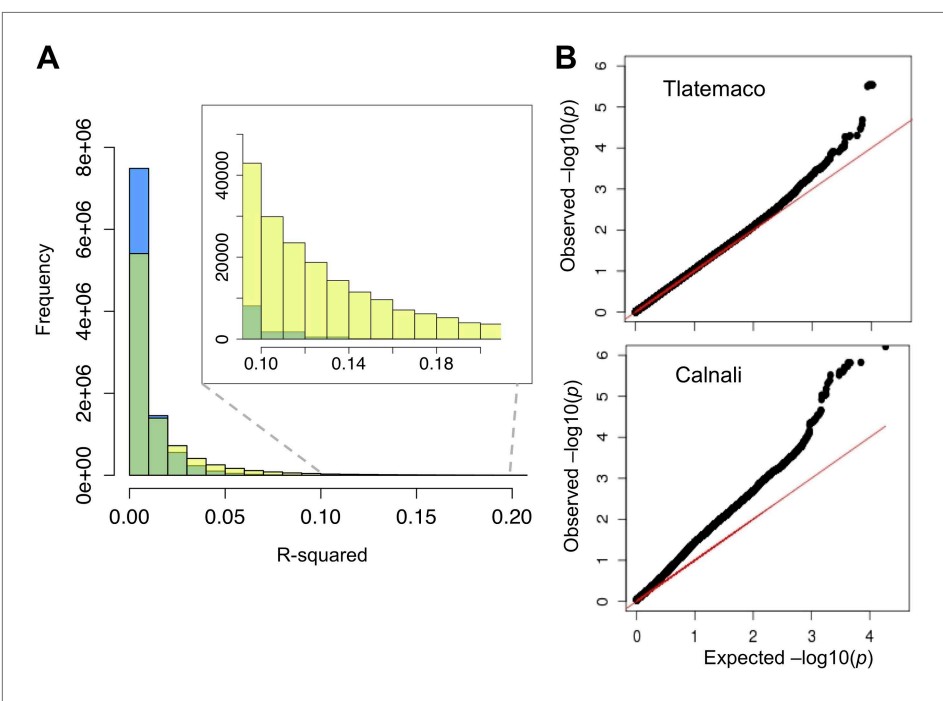

**Figure 2**. $R^2$ distribution and p-value distributions of the sites analyzed in this study. (**A**) Genome-wide distribution of randomly sampled $R^2$ values for markers on separate chromosomes (see *Figure 2—figure supplement 1* for $R^2$ decay by distance; *Figure 2—figure supplement 2* for a genome-wide plot). Blue indicates the distribution in Tlatemaco while yellow indicates the distribution in Calnali. Regions of overlapping density are indicated in green. The average genome-wide $R^2$ in Tlatemaco is 0.003 and in Calnali is 0.006. (**B**) qq-plots of $-\log_{10}$(p-value) for a randomly selected subset of unlinked sites analyzed in this study in each population; expected p-values are drawn from p-values of the permuted data.

The following figure supplements are available for figure 2:

**Figure supplement 1**. Decay in linkage disequilibrium.

**Figure supplement 2**. Genome-wide linkage disequilibrium plot for Tlatemaco.

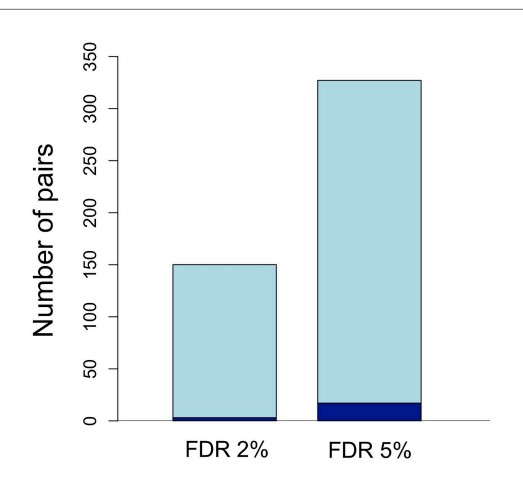

**Figure 3**. Number of unlinked pairs in significant linkage disequilibrium and expected false discovery rates. Plot showing number of pairs of sites in significant LD in both populations in the stringent and relaxed data sets (light blue). The expected number of false positives in each data set is shown in dark blue, and was determined by simulation (see main text; *Figure 3—figure supplement 1*).
The following figure supplements are available for figure 3:

**Figure supplement 1**. False discovery rate (FDR) at different p-value thresholds.

demographic history. The joint distribution of p-values in the two populations implies several hundreds of unlinked locus pairs are in significant LD in both populations (*Figure 3—figure supplement 1*); 327 pairs of regions were significant at FDR = 5%, and 150 pairs were significant at a more stringent FDR = 2% (*Figure 3*). The genomic distribution of loci in significant LD at FDR = 5% is shown in *Figure 4*. For simplicity, we focus on statistics for the less stringent data set (*Supplementary file 1A*), but nearly identical results were found for the more stringent data set (*Tables 1 and 2*).

Average $R^2$ for unlinked regions in LD is 0.08 in Tlatemaco and 0.12 in Calnali, both significantly higher than the genomic background (p<0.001, by bootstrapping). LD regions were non-randomly distributed in the genome ($\chi^2$ = 87, df = 23, p<3e−9). Contrary to findings of a large-X effect in other taxa ('Discussion'), we do not find a significant excess of LD pairs involving the X chromosome (p=0.11, Binomial test). Focusing on the overlap of significant regions in both populations allowed us to narrow candidate regions (*Figure 4—figure supplement 3*). Regions in significant LD in both populations have a median size of 45 kb (*Figure 4—figure supplement 1*; *Figure 4—figure supplement 2*); 67% of regions contain 10 genes or fewer, and 13 pairs of regions are at single gene resolution (*Supplementary file 1B*). Approximately, 10% of the genomic regions identified are very large and contain hundreds of genes (>5 Mb, *Supplementary file 1A*), potentially as a result of reduced recombination or selection. Unlinked regions in significant LD were more strongly linked to neighboring loci (*Figure 4—figure supplement 4*), ruling out the possibility that mis-assemblies underlie the patterns we observe.

## Evidence for hybrid incompatibilities

Models of selection against hybrid genotypes (*Figure 5*, *Figure 5—figure supplement 1*) predict that certain genotype combinations will be less common (*Karlin, 1975*). In particular, selection against hybrid incompatibilities is expected to generate positive R, or conspecific associations between loci. Among loci in significant LD, we found an excess of conspecific associations in both hybrid populations (94% of pairs in Calnali and 67% of pairs in Tlatemaco, p<0.001 for both populations relative to the genomic background by bootstrapping).

We focus all subsequent analyses on the subset of significant LD pairs (FDR = 0.05) with conspecific associations in both populations (207 locus pairs) because these sites will be enriched for hybrid incompatibilities, but similar results are observed for the whole data set (*Supplementary file 1C*). To estimate parameters under a classic BDM incompatibility model (*Figure 5—figure supplement 1*), we use an approximate Bayesian approach to simulate selection on two-locus interactions ('Materials and methods'). Though it is likely that other types of hybrid incompatibilities were identified in our analysis, estimates are similar using other models ('Materials and methods'; *Figure 5—figure supplement 1C*). These simulations demonstrate that our results are well described by a model of selection against hybrid incompatibilities (*Figure 5*, *Figure 5—figure supplement 1*) and that sites are on average under weak to moderate selection (*Figure 5*, maximum a posteriori estimates for Tlatemaco ŝ = 0.027, and Calnali ŝ = 0.074).

## Elevated divergence at loci linked to incompatibilities

Strikingly, the median divergence between *X. birchmanni* and *X. malinche* at loci in significant conspecific LD (FDR = 0.05) was much higher than the genomic background (p<0.001 by bootstrapping,

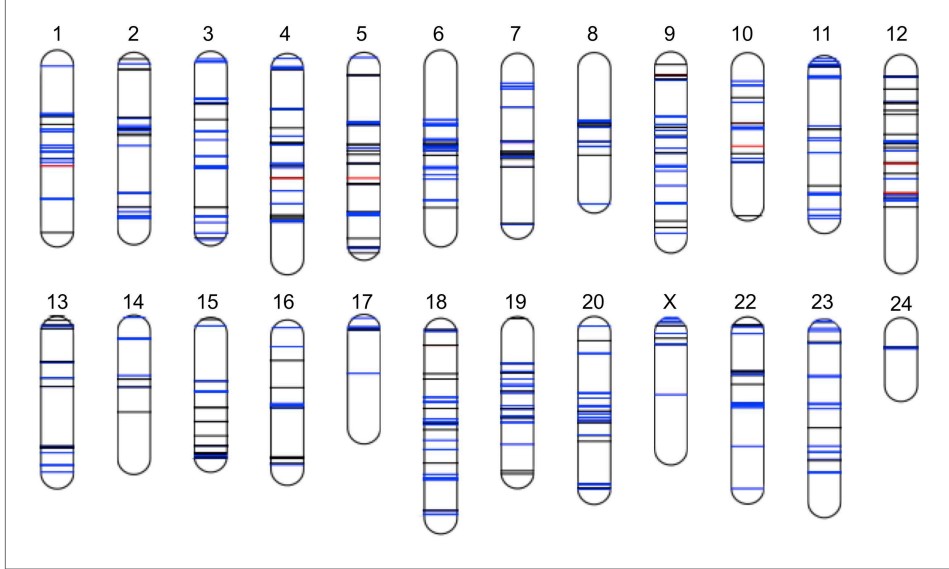

**Figure 4**. Distribution of sites in significant linkage disequilibrium throughout the Xiphophorus genome. Schematic of regions in significant LD in both populations at FDR 5%. Regions in blue indicate regions that are positively associated in both populations (conspecific in association), regions in black indicate associations with different signs of R in the two populations, while regions in red indicate those that are negatively associated in both populations (heterospecific in association). Chromosome lengths and position of LD regions are relative to the length of the assembled sequence for that linkage group; most identified LD regions are <50 kb (*Figure 4—figure supplement 1*; *Figure 4—figure supplement 2*; *Figure 4—figure supplement 3*). Analysis of local LD excludes mis-assemblies as the cause of these patterns (*Figure 4—figure supplement 4*).

The following figure supplements are available for figure 4:

**Figure supplement 1**. Log$_{10}$ distribution of LD region length in base pairs.

**Figure supplement 2**. Plot of the number of recombination breakpoints detected along linkage group 2.

**Figure supplement 3**. Example of the use of data from two populations to narrow candidate regions in cross-chromosomal LD.

**Figure supplement 4**. Regions in cross-chromosomal LD are also in LD with their neighbors.

---

*Figure 6A*). Elevated divergence could be caused by differences in selection on individual loci, differences in mutation rate, or reduced susceptibility of these genomic regions to homogenization by ongoing gene flow. To distinguish among these causes, we examined rates of synonymous substitution (dS). Elevated divergence compared to the genomic background is also observed at synonymous sites (p<0.01 by bootstrapping, *Table 2*). We also examined the same regions in two swordtail species in an independent *Xiphophorus* lineage, *X. hellerii* and *X. clemenciae* and found that the level of genomic divergence in this species pair was not significantly different from the genomic background (*Figure 6B*). Together, these results imply that variation in selective constraint or mutation rate do not explain elevated divergence between *X. birchmanni* and *X. malinche* at loci in conspecific LD.

## Gene ontology analysis

We performed gene ontology (GO) analysis on unlinked regions in significant conspecific LD and, surprisingly, found no significantly enriched GO categories ('Materials and methods'). This result holds when restricting the analysis to regions resolved to only a few genes (≤10 genes, 242 regions). This suggests that regions in significant LD contain genes with a broad range of functional roles.

**Table 1.** Comparison of results for sites in significant LD at two different p-value thresholds

| Data set | Number of pairs | Proportion of pairs with conspecific associations |
|---|---|---|
| Stringent (FDR<2%) | 150 | Tlatemaco: 72% (p<0.001) Calnali: 94% (p<0.001) |
| Relaxed (FDR 5%) | 327 | Tlatemaco: 67% (p<0.001) Calnali: 94% (p<0.001) |

p-values were determined resampling the genomic background, see main text for details.

## Discussion

In this study, we assign ancestry to nearly 500,000 markers genome-wide in samples from two hybrid fish populations, providing a portrait of the genetic architecture of hybrid incompatibilities between two closely related species at unprecedented resolution. We discover significant LD between hundreds of pairs of unlinked genomic regions and show that a model of selection against hybrid incompatibilities describes the observed conspecific LD patterns. This implies that many negative epistatic interactions segregate in these hybrid fish populations despite the fact that intrinsic post-zygotic isolation is thought to be weak (*Rosenthal et al., 2003*).

## How many regions are involved in hybrid incompatibility?

Our analysis focuses on a high confidence set of unlinked loci in significant LD. Despite these findings, the true number of loci involved in incompatibilities may involve hundreds more interactions for several reasons. First, we conservatively exclude interactions within chromosomes due to the lack of detailed genetic map information. Second, we have only moderate power to detect incompatibilities (e.g., ~30% using parameter estimates for Calnali). Third, relaxing the p-value threshold suggests the true number of pairs in significant LD could be much larger (*Figure 3—figure supplement 1*). Finally, our experimental design incorporates two populations with opposite trends in genome-wide ancestry and independent histories of hybridization. This conservative approach allows us to exclude effects of population history, but false negatives in our joint analysis may result from effects that are population-specific (for example, different effects of extrinsic selection in the two populations). Investigating the role of such a large number of hybrid incompatibilities in determining the structure of hybrid genomes is an exciting area for future empirical and theoretical research.

## How strong is selection on hybrid incompatibilities?

Using an approximate Bayesian approach, we estimate that the average selection coefficient on negative epistatic interactions is in the vicinity of $s = 0.03$ (Tlatemaco) to $s = 0.07$ (Calnali); scaling these by estimates of the effective population size ($2 Ns \sim 10$ and $\sim 45$, respectively) suggests that selection is strong enough to be deterministic but of moderate strength relative to drift. Given the large number of putative incompatibilities, even weak selection would introduce substantial genetic load in hybrids. Depending on dominance effects, this could explain why hybrid populations are skewed in genome-wide ancestry. However, we may overestimate the potential for genetic load if epistatic interactions are complex. Though this study focuses on pairwise comparisons because statistically evaluating high-order interactions in a data set of this size is intractable, more complex interactions are predicted to be likely (*Orr, 1995*). The fact that many locus pairs (~40% of regions localized to 1 Mb or less) identified in this study interact with multiple partners provides indirect support for this prediction.

**Table 2.** Sites in significant LD are more divergent per site than the genomic background

| Mutation type | Median divergence genomic background | Median divergence stringent | Median divergence relaxed |
|---|---|---|---|
| All sites | 0.0040 | 0.0045 (p<0.001) | 0.0044 (p<0.001) |
| Nonsynonymous | 0.00040 | 0.00065 (p=0.001) | 0.00040 (p=0.6) |
| Synonymous | 0.0040 | 0.0048 (p<0.001) | 0.0045 (p=0.004) |

Results shown here are limited to regions that had conspecific associations in both populations (stringent dataset: 200 regions, relaxed dataset: 414 regions). p-values were determined by resampling the genomic background.

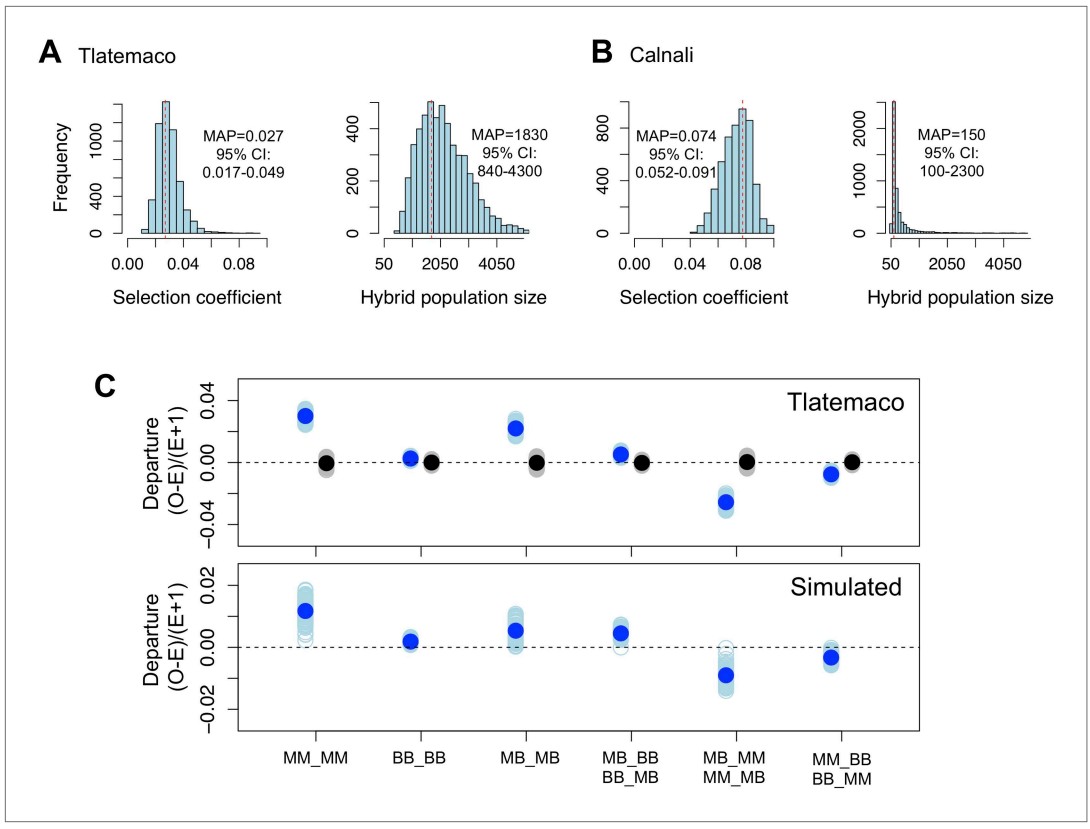

**Figure 5**. Loci in significant conspecific linkage disequilibrium show patterns consistent with selection against hybrid incompatibilities. (**A**) Posterior distributions of the selection coefficient and hybrid population size from ABC simulations for Tlatemaco and (**B**) Calnali. The range of the x-axis indicates the range of the prior distribution, maximum a posteriori estimates (MAP) and 95% CI are indicated in the inset. (**C**) Departures from expectations under random mating in the actual data (top—blue points indicate LD pairs, black points indicate random pairs from the genomic background) and samples generated by posterior predictive simulations (bottom, see 'Materials and methods'). The mean is indicated by a dark blue point; in the real data (top) smears denote the distribution of means for 1000 simulations while in the simulated data (bottom) smears indicate results of each simulation. Genotypes with the same predicted deviations on average under the BDM model have been collapsed (***Figure 5—figure supplement 1***, but see ***Figure 5—figure supplement 3***) and are abbreviated in the format locus1_locus2. These simulations show that the observed deviations are expected under the BDM model. The posterior distributions for *s* and hybrid population size are correlated at low population sizes (***Figure 5—figure supplement 2***). Deviations in Calnali also follow expectations under the BDM model (***Figure 5—figure supplement 3***).

The following figure supplements are available for figure 5:

**Figure supplement 1**. Different fitness matrices associated with selection against hybrid incompatibilities.

**Figure supplement 2**. Joint posterior distribution of hybrid population size and selection coefficient.

**Figure supplement 3**. Deviations in genotype combinations compared to expected values under a two-locus selection model in both populations.

## Insights from a genome-wide approach in natural hybrids

Most work on hybrid incompatibilities has focused on characterizing specific incompatibilities at candidate genes, such as those associated with mapped QTL distinguishing species (***Ting et al., 1998***; ***Presgraves et al., 2003***; ***Lee et al., 2008***; ***Tang and Presgraves, 2009***; ***Moyle and Nakazato, 2010***). The idea that negative epistatic interactions may be pervasive in closely related species or populations has come from multiple studies of potential candidate genes (e.g., *Arabidopsis thaliana*: ***Bomblies et al., 2007***; ***Smith et al., 2011***; *Xiphophorus*: ***Nairn et al., 1996***; *Drosophila*: ***Barbash et al., 2000***;

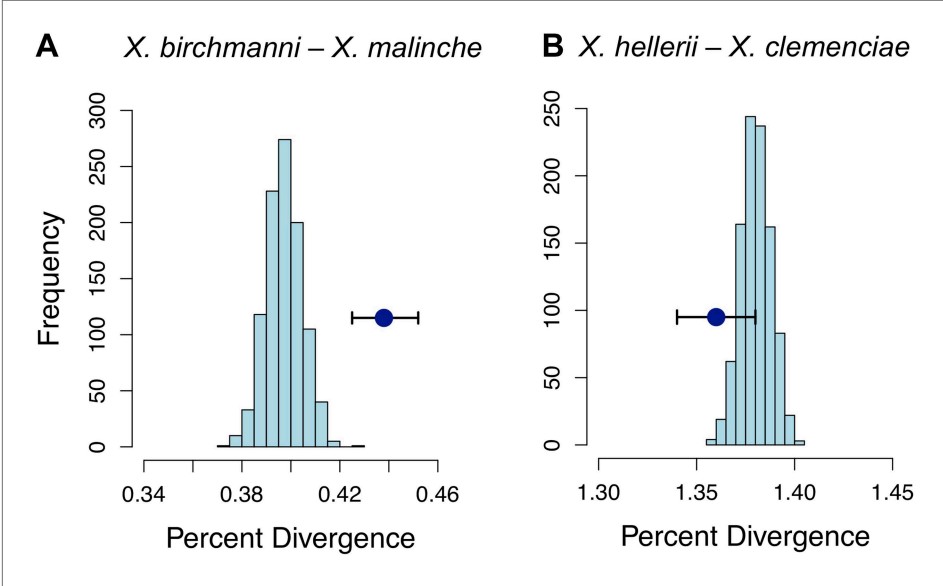

**Figure 6**. Divergence of LD pairs compared to the genomic background in two species comparisons. (**A**) Regions identified in *X. birchmanni* and *X. malinche* and (**B**) orthologous regions in *X. hellerii* and *X. clemenciae*. The blue point shows the average divergence for genomic regions within significant LD pairs, and whiskers denote a 95% confidence interval estimated by resampling genomic regions with replacement. The histogram shows the distribution of the average divergence for 1000 null data sets generated by resampling the genomic background without replacement.

*Bayes and Malik, 2009*; *Tang and Presgraves, 2009*; *Caenorhabditis elegans: Seidel et al., 2008*) and several genome-wide studies (*Harushima et al., 2001*; *Presgraves, 2003*; *Payseur and Hoekstra, 2005*; *Moyle and Graham, 2006*; *Masly and Presgraves, 2007*; *Matute et al., 2010*). These studies suggest that on the order of 100 incompatibilities explain isolation between closely related species. In contrast, though *X. malinche* and *X. birchmanni* have a similar divergence time (~2 $N_e$ generations) to a previously studied *Drosophila* species pair (*Masly and Presgraves, 2007*), we identify approximately fourfold more genetic incompatibilities at FDR = 0.05.

What accounts for this difference? Previous studies on the genomic distribution of hybrid incompatibilities have focused almost entirely on incompatibilities involved in post-zygotic isolation (in some studies, exclusively in males; *Presgraves, 2003*). However, such studies can only provide a lower limit on the number of loci involved in reproductive isolation between lineages. A recent study in *Drosophila simulans* and *D. sechellia* investigated the effects of interspecific competition on the fitness of introgressed lines (*Fang et al., 2012*). Though introgressed lines had no detectable differences in fertility or viability, the authors nonetheless detected strong selection on hybrids in competition experiments (*Fang et al., 2012*), implying that the number of loci involved in reproductive isolation has been vastly underestimated in most studies.

Our results lend support to this point of view. The hybrid genomes we analyze in this study have been exposed to over 30 generations of intrinsic and extrinsic selection. Given that post-zygotic isolation between *X. birchmanni* and *X. malinche* is weak (*Rosenthal et al., 2003*), we propose that extrinsic selection is more likely the cause of the majority of hybrid incompatibilities detected in this study. Future research comparing hybrid incompatibilities in lab hybrids to natural hybrids will begin to elucidate how many incompatibilities are targets of extrinsic vs intrinsic selection.

Many studies of hybrid incompatibilities have focused on organisms with clear species boundaries— those that do not frequently hybridize in nature. In species that do frequently hybridize, early research supported the conclusion that these species retain their identity through a few highly differentiated genomic regions, inferred to contain genes responsible for reproductive isolation (*Turner et al., 2005*; *Ellegren et al., 2012*). More recent studies have suggested that even hybridizing species remain genetically differentiated through much of the genome (*Lawniczak et al., 2010*; *Michel et al., 2010*).

Our findings in *X. birchmanni* and *X. malinche* support the latter conclusion and suggest that hybrid incompatibilities may be a common mechanism of restricting gene flow genome-wide even in species with incomplete reproductive isolation.

## Evidence for reduced gene flow associated with putative incompatibilities

Theory predicts that more rapidly evolving genomic regions will be more likely to result in BDM incompatibilities (*Orr, 1995*; *Orr and Turelli, 2001*). However, an issue that arises in testing this prediction is that functionally diverged regions associated with hybrid incompatibilities may also resist homogenization due to gene flow, conflating the cause and the effect of elevated divergence (but see below). While our study confirms that unlinked genomic regions in LD pairs are significantly more diverged between *X. birchmanni* and *X. malinche* compared to the genomic background (*Figure 6*), we also find that these regions do not show elevated divergence in an independent comparison of *Xiphophorus* fish (*Figure 6B*). This supports the hypothesis that these regions are resistant to homogenization due to ongoing gene flow between *X. birchmanni* and *X. malinche*. This finding is interesting because theoretical work suggests BDM incompatibilities are ineffective barriers to gene flow, especially when migration rates are high (*Gavrilets, 1997*; *Gompert et al., 2012*), but incompatibilities in which all hybrid genotypes are under selection more effectively limit gene flow (*Gavrilets, 1997*).

## Functional evaluation of loci associated with putative incompatibilities

Remarkably, we found not a single significantly enriched GO category in well-resolved pairs in significant LD. This is in stark contrast to previous studies, such as that of *Payseur and Hoekstra (2005)*, who found 17 over-represented GO categories in a data set of ~180 pairs of loci (at 2 Mb resolution) detected in *Mus*. Our study suggests a much more equal representation of functional categories among genes involved in incompatibilities.

One of the first putative BDM incompatibilities identified at the genetic level involves the *Xmrk-2* gene in *Xiphophorus* hybrids. Hybrids between many *Xiphophorus* species develop lethal melanomas which have been hypothesized to reinforce species boundaries (*Orr and Presgraves, 2000*). Decoupling of *Xmrk-2* from its repressor (thought to be the gene *cdkn2x*) through hybridization triggers melanoma development (*Nairn et al., 1996*). Though melanomas can develop in *X. malinche*–*X. birchmanni* hybrids, we found no evidence of LD between *Xmrk-2* and *cdkn2x* in either population. This may support previous hypotheses that melanoma is not under strong selection in hybrids because it affects older individuals (*Schartl, 2008*) or provides an advantage in mate choice (*Fernandez and Morris, 2008*; *Fernandez and Bowser, 2010*). Alternatively, *cdkn2x* may not in fact be the repressor of *Xmrk-2*.

## No evidence for a large X effect

Theory predicts that the X-chromosome will play a major role in the establishment of reproductive isolation due to Haldane's rule, faster-X evolution or meiotic drive (*Presgraves, 2008*). Intriguingly, we do not see an excess of interactions involving group 21, the putative X chromosome (*Schartl et al., 2013*). This is in contrast to results in a large number of species that demonstrate that sex chromosomes play a disproportionate role in the evolution of reproductive isolation (*Sperling, 1994*; *Presgraves, 2002*; *Payseur et al., 2004*; *Turner et al., 2005*; *Pryke, 2010*), including studies on LD (*Payseur and Hoekstra, 2005*). However, the X chromosome in *Xiphophorus* is very young (*Schartl, 2004*) and sex determination may also be influenced by autosomal factors (*Kallman, 1984*). Since the non-recombining portion of the Y chromosome is small in *Xiphophorus*, this will reduce the effects of recessive X-chromosome incompatibilities on male fitness.

## What explains significant heterospecific associations?

We detect an excess of conspecific associations among significant LD pairs, which we evaluate (above) in the context of selection on hybrid incompatibilities. However, the proportion of locus pairs in significant LD that are in conspecific association differs dramatically in the two populations. 6% of locus pairs in Calnali and 33% in Tlatemaco have significant heterospecific associations (i.e., significantly negative R) at FDR = 0.05. Heterospecific associations may be the result of beneficial epistatic interactions that result in hybrid vigor. For example, hybrid males are better at buffering the locomotor costs of sexual ornamentation (*Johnson, 2013*) and have an advantage compared to parental males from sexual selection (*Culumber et al., in press*), which could in part counteract the negative fitness effects

of genetic incompatibilities. However, the fact that few loci are heterospecific in association in both populations (2%, *Figure 3—figure supplement 1B*) suggests that these patterns are not repeatable across populations. One possible explanation for this is divergent effects of mate preferences in the two populations. Behavioral studies have shown that *X. malinche* females prefer unfamiliar male phenotypes (*Verzijden et al., 2012*) while *X. birchmanni* females prefer familiar male phenotypes (*Verzijden and Rosenthal, 2011*; *Verzijden et al., 2012*). Given that Tlatemaco hybrids are primarily *malinche* and Calnali hybrids are primarily *birchmanni*, divergent effects of male phenotypes on mating preferences could produce the observed patterns.

## Conclusions

We find hundreds of pairs of unlinked regions in significant LD across the genomes of *X. birchmanni–X. malinche* hybrids in two independent hybrid populations. These associations are largely well described by a model of selection against hybrid incompatibilities, implying that reproductive isolation in these recently diverged species involves many loci. These regions were also more divergent between species than the genomic background, likely as a result of reduced permeability to ongoing gene flow between the species. By using samples from two populations with independent histories of hybridization, we are able to exclude population structure and drift as potential causes of these patterns. Our results suggest that past research has vastly underestimated the number of regions responsible for reproductive isolation between species by focusing on intrinsic postzygotic reproductive isolation. In addition, our results demonstrate that even in species without strong intrinsic post-zygotic isolation, hybrid incompatibilities are pervasive and play a major role in shaping the structure of hybrid genomes.

## Materials and methods

### Genome sequencing and pseudogenome assembly

We created 'pseudogenomes' of *X. malinche* and *X. birchmanni* based on the *X. maculatus* genome reference sequence. As raw materials, we used previously collected Illumina sequence data (Acc # SRX201248; SRX246515) derived from a single wild-caught male for each species (*Cui et al., 2013*; *Schumer et al., 2013*) and the current genome assembly for *X. maculatus* (*Schartl et al., 2013*). We used a custom python script to trim reads to remove low quality bases (Phred quality score <20) and reads with fewer than 30 nucleotides of contiguous high quality bases and aligned these reads to the *X. maculatus* reference using STAMPY v1.0.17 (*Lunter and Goodson, 2011*) with default parameters except for expected divergence set to 0.03. Between 98–99% of reads from both species mapped to the *X. maculatus* reference. Mapped reads were analyzed for variant sites using the samtools/bcftools pipeline (*Li and Durbin, 2009*). We used the VCF files and a custom python script to create a new version of the *X. maculatus* reference sequence for each species that incorporated variant sites and masked any sites that had depth <10 reads or were called as heterozygous.

As an additional step to mask polymorphisms, we prepared multiplexed shotgun genotyping (MSG) libraries (*Andolfatto et al., 2011*) for 60 parental individuals of each species (*Figure 1—figure supplement 1*), generating 78,881,136 single end 101 nucleotide reads at MseI sites for *X. malinche* and 80,189,844 single end 101 nucleotide reads for *X. birchmanni*. We trimmed these reads as described above and mapped them to the *X. malinche* and *X. birchmanni* reference pseudogenomes, respectively, using bwa (*Li and Durbin, 2009*). We analyzed variant sites as described above and excluded all sites that were either polymorphic in the sampled parentals or had fixed differences between the sampled parentals and the reference (excluding indels). After masking, 0.4% of sites genome-wide were ancestry informative markers (AIMs) between *X. birchmanni* and *X. malinche*. The total number of AIMs in the assembled 24 linkage groups was 2,189,807. For the same 60 parental individuals of each species, we evaluated MSG output to determine any markers that did not perform well in genotyping the parental individuals (average probability of matching same-parent <0.9). We found that 1.7% of covered markers performed poorly in *X. malinche* and 0.3% of markers performed poorly in *X. birchmanni*; we excluded these 10,877 markers in downstream analysis, leaving 2,178,930 AIMs.

### Sample collection

The procedures used in this study were approved by the Institutional Animal Care and Use Committee at Texas A&M University (Protocols # 2010-111 and 2012-164). Individuals were collected from two independent hybrid zones (Calnali-mid and Tlatemaco, *Culumber et al., 2011*) in 2009, and between 2012 and 2013. Individuals were caught in the wild using baited minnow traps, and lightly anesthetized

with tricaine methanesulfate. Fin clips were stored in 95% ethanol until extraction. Population turnover rate is high between years, and sites were sampled only once per year. We also performed relatedness analyses to ensure that individuals had not been resampled (data not shown).

## MSG library preparation and sequencing

DNA was extracted from fin clips using the Agencourt bead-based purification method (Beckman Coulter Inc., Brea, CA) following manufacturer's instructions with slight modifications. Fin clips were incubated in a 55°C shaking incubator (100 rpm) overnight in 94 µl of lysis buffer with 3.5 µl 40 mg/ml proteinase K and 2.5 DTT, followed by bead binding and purification. Genomic DNA was quantified using a Typhoon 9380 (Amersham Biosciences, Pittsburgh, PA) and evaluated for purity using a Nanodrop 1000 (Thermo Scientific, Wilmington, DE); samples were diluted to 10 ng/µl.

MSG libraries were made as previously described (*Andolfatto et al., 2011*). Briefly, 50 ng of DNA was digested with MseI; following digestion custom barcodes were ligated to each sample. 5 µl of sodium acetate and 50 µl of isopropanol were added to each sample and samples were pooled (in groups of 48) and precipitated overnight at −20°C. Following overnight precipitation, samples were extracted and resuspended in TE (pH 8.0) and purified through a phenol–chloroform extraction and Agencourt bead purification. Pooled samples were run on a 2% agarose gel and fragments between 250 and 500 bp were selected and purified. 2 ng of each pooled sample was amplified for 14–16 PCR cycles with custom indexed primers allowing us to pool ~180 samples for sequencing on one Illumina lane. Due to multiplexing with other libraries, samples were sequenced on a total of four Illumina HiSeq 2000 lanes with v3 chemistry. All raw data are available through the NCBI Sequence Read Archive (SRA Accession: SRX544941).

Raw reads were parsed by index and barcode; the number of reads per individual ranged from 0.4 to 2.8 million reads, with a median of 900,000 reads. After parsing, 101 bp reads were trimmed to remove low quality bases (Phred quality score <20) and reads with fewer than 30 bp of contiguous high quality bases. If an individual had more than 2 million reads, reads in excess of 2 million were excluded to improve the speed of the MSG analysis pipeline.

## MSG analysis pipeline

The following parameters were specified in the MSG configuration file: recombination rate recRate = 240, rfac = 3, *X. birchmanni* error (deltapar1) = 0.05, *X. malinche* error (deltapar2) = 0.04. See 'Materials and methods' for details on parameters and parameter determination. All individuals were initially analyzed with naive priors (probability of ancestry for parent 1 = 0.33, parent 2 = 0.33, and heterozygous = 0.33) with the MSG v0.2 pipeline (https://github.com/JaneliaSciComp/msg). Based on genome-wide ancestry proportions given these priors, population-specific priors were calculated for Tlatemaco (homozygous *X. malinche* = 0.49, heterozygous = 0.42, homozygous *X. birchmanni* = 0.09) and Calnali (homozygous *X. malinche* = 0.04, heterozygous = 0.32, homozygous *X. birchmanni* = 0.64). These estimates were used as new priors for a subsequent run of the MSG pipeline. This resulted in genotype information at 1,179,187 ancestry informative markers (~50% of the total number of ancestry informative markers). MSG ancestry posterior probability files were thinned to exclude markers that were missing or ambiguous in >15% of individuals leaving 469,400 markers (~820 markers/Mb). Following this initial culling, markers were further thinned using a custom Python script to exclude adjacent markers that did not differ in posterior probability values by ±0.1. This resulted in 12,269 markers for linkage disequilibrium analysis; the median distance between thinned markers was 2 kb (mean = 48 kb). Individuals were considered hybrids if at least 10% of their genome was contributed by each parent; using this definition, 100% of individuals collected from Tlatemaco and 55% of individuals collected from Calnali were hybrids. Three more individuals from Calnali were excluded because their hybrid index was not within the 99% CI of the mean hybrid index. Including them resulted in a significant deviation from the expected $R^2$ between unlinked sites of 1/2n (n: number of sampled individuals).

## Estimates of hybrid zone age

The expected number of generations since initial hybridization was estimated using the LD decay with distance method described in *Hellenthal et al. (2014)*. First, we used genome sequences of 5 *Xiphophorus* outgroups (*X. maculatus, X. hellerii, X. clemenciae, X. variatus,* and *X. nezahualcoyotl*) to identify autapomorphic loci in *X. malinche* and *X. birchmanni* respectively. We limit the analyses to only the autapomorphic loci for the minor parental species in each hybrid zone. We then fit an exponential curve D = a*exp (−T*x), where 'D' is disequilibrium, 'a' is a coefficient, 'T' is time since hybridization in

generations and 'x' is the physical distance between markers in Morgans (scripts are available in the Dryad data repository under DOI: 10.5061/dryad.q6qn0, *Schumer et al., 2014*). Because we do not have access to a recombination map for our species, we assumed a uniform recombination rate of 1 cM/378 kb (*Walter et al., 2004*). This assumption can underestimate the time since hybridization in some cases (*Sankararaman et al., 2012*), but better estimates await more detailed genetic map information.

## Quantifying LD and establishing significance

For each pairwise combination of markers (*Figure 2—figure supplement 2*), we used a custom php script to calculate R, the correlation coefficient. R is typically defined as D, the disequilibrium coefficient, scaled by the square root of the product of the allele frequencies at the two loci (*Hartl and Clark, 2007*). We use the methods outlined in *Rogers and Huff (2009)* for calculation of R using unphased data. We recorded whether R was positive or negative, corresponding to conspecific vs heterospecific association. To assess significance of correlations, we used a Bayesian ordered logistic regression as implemented in the R package bayespolr (http://rss.acs.unt.edu/Rdoc/library/arm/html/bayespolr.html) to estimate Student's t; we used this estimate to determine the two-sided p-value for the correlation. We only considered pairs of markers belonging to different linkage groups; intrachromosomal comparisons were excluded due to concerns about false positives caused by recombination rate variation (scripts are available in the Dryad data repository under DOI: 10.5061/dryad.q6qn0).

To determine our expected false discovery rates (FDRs) associated with given p-value significance thresholds, we used a simulation approach. For LD analysis, we surveyed 12,269 markers (reduced from 1.2 million, see above), but many of these markers are tightly clustered. We used the Matrix Spectral Decomposition method described in *Li and Ji (2005)* as implemented in the program matSpDlite (*Nyholt, 2004*), to determine the effective number of markers. We used the correlation matrix for each pair-wise marker from Tlatemaco for these calculations; calculations based on the correlation matrix from Calnali resulted in a similar but slightly lower number of tests. We determined based on this analysis that the effective number of markers is 1087. Based on these results, we randomly selected 1087 markers from our data set, randomly shuffled genotypes within columns, calculated $R^2$ and p-values for the entire data set, and determined the expected number of false positives at different p-value thresholds. We repeated this procedure 1000 times. We compared the average number of false positives to the total number of positives in the actual data set at a number of p-value thresholds. Based on this analysis, we determined that $p<0.013$ in both populations resulted in an expected false discovery rate (FDR) of 0.05 for 1087 independent markers (excluding within chromosome comparisons), while $p<0.007$ resulted in an expected FDR of 0.02. Our analyses focused on the FDR = 0.05 data set, but we repeated these analyses with a more restricted data set (FDR = 0.02, *Tables 1 and 2*). We also performed simulations to investigate the potential effects of demographic processes on p-value distributions (see below).

## Establishing the number of independent LD pairs

In most cases, dozens to hundreds of contiguous markers showed the same patterns of LD. To cluster these markers and delineate between independent and non-independent LD blocks, we used an approach designed to conservatively estimate the number of LD pairs. Within adjacent clusters on the same chromosome, we tested for independence between clusters of sites by determining the p-value for $R^2$ between a focal site and the last site of the previous LD cluster. If $p>0.013$ (our FDR = 0.05 significance threshold), we counted the focal site as the first site in a new cluster.

## Excluding mis-assemblies as causes of long range LD

If regions of the *Xiphophorus* genome are misassembled, incorrect assignment of contigs to different linkage groups could generate strong cross-chromosomal linkage disequilibrium (see for e.g., *Andolfatto et al., 2011*). To evaluate this possibility, we focused on markers at the edges of identified LD blocks and examine patterns of local LD in these regions (*Figure 4—figure supplement 4*). If markers had neighboring markers within 300 kb (86%), we evaluated whether the marker had stronger LD with neighboring markers than detected in any cross-chromosomal comparisons. Only 1.5% of markers in Calnali and 0.6% in Tlatemaco had stronger cross-chromosomal LD than local LD.

## Analysis of potential hybrid incompatibilities

Selection against hybrid incompatibilities is expected to generate an excess of conspecific associations. To investigate whether regions in significant LD were more likely to have conspecific associations, we

determined the direction of association between markers in each population. We compared the sign of R in pairs in significant LD (327 pairs at FDR = 0.05) to the sign of R in 1000 data sets of the same size composed of randomly selected pairs from the genomic background (p>0.013 in each population).

## Simulations of selection on hybrid incompatibilities

To investigate what levels of selection might be required to generate the patterns we observe, we use a model of selection on locus pairs following *Karlin (1975)* and a regression approach to approximate Bayesian inference using summary statistics as implemented in the program ABCreg (*Beaumont et al., 2002*; *Jensen et al., 2008*; *Thornton 2009*). We focus only on sites that have positive R (207 pairs) since these sites are expected to be enriched for hybrid incompatibilities.

Because selection on two-locus interactions results in changes in the frequency of particular genotypes, we used the frequency of double homozygous genotypes for the major parent (Tlatemaco: MM_MM, Calnali: BB_BB), frequency of homozygous–heterozygous genotypes for the minor parent (Tlatemaco: MB_BB and BB_MB, Calnali: MM_MB and MB_MM), and average final ancestry proportion as summary statistics. Under the BDM incompatibility model, two distinct fitness matrices are possible (*Figure 5—figure supplement 1*). Because these models are equally likely (*Coyne and Orr, 2004*), we used the random binomial function in R to assign the 207 conspecific-associated locus pairs to each fitness matrix for each simulation. To simplify our simulations, we assume that selection is equal on all genotype combinations that have not previously been exposed to selection in ancestral populations (see *Figure 5—figure supplement 1*). For each simulation, we drew from uniform prior distributions for 4 parameters. Limits on the prior distribution for admixture proportions for the two populations were determined as 0.5 to A, where A is the 95% CI of 1000 bootstrap resamplings of population ancestry from the observed data. Each simulated replicate was generated as follows:

1. Draw values for $s$ (0–0.1), initial admixture proportions (Tlatemaco 0.5–0.72, Calnali 0.18–0.5), number of generations of selection (Tlatemaco 40–70, Calnali 20–50), and hybrid population size (50–5000).
2. Random assignment of 207 pairs to each of two possible BDM incompatibility fitness matrices.
3. Calculate expected frequencies of each two-locus genotype using these priors and the methods described by *Karlin (1975)*, introducing drift at each generation as sampling of 2N gametes.
4. After iterating through step 3 for t generations, we multinomially sampled expected frequencies from step 3 for n individuals. To account for variation in sample size, we simulated the actual distribution of sample sizes in the observed data.
5. Calculate the mean of each summary statistic.
6. Repeat for 1,000,000 simulations.
7. Run ABCreg with a tolerance of 0.5%.

These simulations produced well-resolved estimates of the selection coefficient, $s$, and hybrid population size, N (*Figure 5*). We also repeated these simulations using a model of selection against all hybrid genotypes (*Figure 5—figure supplement 1C*). These simulations also resulted in well-resolved posterior distributions of $s$ and N and similar maximum a posteriori (MAP) estimates for both populations (Tlatemaco s = 0.015, N = 4360; Calnali s = 0.043, N = 270). This model may be more consistent with incompatibilities arising from co-evolving loci or selection against hybrid phenotypes. Scripts for this analysis are available in the Dryad data repository under DOI: 10.5061/dryad.q6qn0.

To check the consistency of our simulations with the observed data, we performed posterior predictive simulations by randomly drawing 100 values from the joint posterior (of N, s, generations of selection, and admixture proportions) with replacement for each population. For each draw, we then simulated selection using these parameters, applying the same significance threshold as we applied to the real data, until 207 pairs had been generated. Departures from expectations under random mating were compared to departures in the real data (*Figure 5*, *Figure 5—figure supplement 3*).

## Genome divergence analyses

Regions involved in hybrid incompatibilities are predicted to be more divergent than other regions of the genome for a number of reasons (see main text). To evaluate levels of divergence relative to the genomic background, we compared divergence (calculated as number of divergent sites/length of region) between *X. malinche* and *X. birchmanni* at 207 regions in significant conspecific LD (FDR = 0.05) in both populations to 1000 data sets of the same size generated by randomly sampling regions throughout the genome that were not in significant LD (p>0.013 with all unlinked regions)

using a custom perl script and the program fastahack (https://github.com/ekg/fastahack). For LD regions that included only 1 marker (n = 60), we included the flanking region defined by the closest 5′ and 3′ marker. To analyze coding regions, we extracted exons from these regions and calculated dN, N, dS, and S for each gene using codeml in PAML with the F3x4 codon model (*Yang, 1997*; scripts are available in the Dryad data repository under DOI: 10.5061/dryad.q6qn0). For a phylogenetically independent comparison, we repeated this analysis using pseudogenomes for two swordtail species for which we previously collected genome sequence data, *X. hellerii* and *X. clemenciae* (*Schumer et al., 2013*). Repeating all analyses for the full data set (i.e., including pairs in heterospecific LD in one or both populations) did not substantially change the results (*Supplementary file 1C*).

## Gene ontology analysis of genomic regions in conspecific LD

To determine whether there is significant enrichment of certain gene classes in our data set, we annotated regions in significant LD. We only considered LD regions that contained 10 genes or fewer to limit our analysis to regions that are reasonably well-resolved. After excluding regions with no genes, this resulted in 242 regions for analysis containing 202 unique genes. We used the ensembl annotation of the *X. maculatus* genome (http://www.ensembl.org/Xiphophorus_maculatus) to identify the HUGO Genome Nomenclature Committee (HGCN) abbreviation for all the genes in each region. Using the GOstats package in R, we built a custom *Xiphophorus* database using the HGCN gene names listed in the genome and matching each of these to Gene Ontology (GO) categories as specified in the annotated human genome database (in bioconductor 'org.Hs.eg.db'). This resulted in a total of 12,815 genes that could be matched to GO categories. We then tested for functional enrichment in GO categories, using the GOstats and GSEABase packages in R and a p-value threshold of <0.01.

## Modeling the effect of demographic processes on LD

Demographic processes such as bottlenecks and continued migration can affect LD measures and could potentially increase our false discovery rate. To explore how demographic changes might influence LD p-value distributions, we performed coalescent simulations using the MAP estimate of hybrid population size (co-estimated with other parameters using ABC, see above).

We used Hudson's *ms* (*Hudson, 1990*) to simulate an unlinked pair of regions in two populations. We calculated time of admixture relative to the time of speciation using the relationship $Tdiv_{4N}=(1/2)((D_{xy}/\theta) -1)$, where $D_{xy}$ is the average number of substitutions per site between species; we previously estimated $\rho$, the population recombination rate, $\theta$, the population mutation rate ($\rho =\theta = 0.0016$ per site), and $N_e$, the effective population size ($N_e = 10,500$), for *X. birchmanni* based on the whole genome sequences (*Schumer et al., 2013*). Because parameter estimates were similar for *X. malinche*, for simplicity, we used these estimates for both parental populations. For population 1, we set the time of admixture ($t_{admix/4N}$) to 0.0014 generations, the proportion *X. malinche* to 0.7 (derived from the average hybrid index in samples from Tlatemaco), and the sample size to 170 diploids. For population 2, we set $t_{admix/4N}$ to 0.000875, the proportion *X. malinche* to 0.2 (derived from the average hybrid index in samples from Calnali), and the sample size to 143 diploids. We specified a bottleneck at $t_{admix/4N}$ reducing the population to 2% its original size in simulations of Calnali and 18% its original size in simulations of Tlatemaco (based on results of ABC simulations, see 'Results'). In each simulated replicate, we selected one substitution from each unlinked region and accepted a pair of sites if they were fixed for different states between parents (evaluated by generating 30 chromosomes of each species per simulation). Simulations were performed until 100,000 pairs had been simulated (scripts are available in the Dryad data repository under DOI: 10.5061/dryad.q6qn0); we used the rate of false positives from these simulations to calculate the total expected number of false positives given our effective number of tests. Based on these simulations, our expected FDR at p<0.013 is ~10%, slightly larger than our expected FDR based on permutation of the data.

Because we do not have information about the migration history of these populations, we use simulations of migration only to explore how continued migration or multiple admixture events might affect our false discovery rate. We simulate unidirectional migration of individuals from each parental population to the hybrid population per generation (4Nm = 80 for each parental population). Under this migration scenario, our expected FDR at p<0.013 is 15%. In addition to scenarios of ongoing migration, we simulated migration bursts. For a short time interval that corresponds to ~1 generation starting ~10 generations ago ($t_{mig/4N} = 0.00025–0.000275$), we set the migration rate to 4Nm = 4000, or 10% of the population made up of migrants. We simulated three scenarios: (1) migration from the

major parent (expected FDR 15%), (2) migration from the minor parent (expected FDR 11%), and (3) migration from both parents (4Nm = 2000 for each parental population, expected FDR 11%). None of these demographic scenarios increased expected FDR above 15%.

## MSG parameter determination and power simulations

Because MSG has not previously been used to analyze natural hybrids, we evaluated performance at a range of parameters and performed power simulations. To optimize the Hidden Markov Model (HMM) parameters of MSG for analysis of natural hybrids, we used a combination of empirical data and simulations. We set the error rate parameter (deltapar) for each parent based on the genome wide average rate of calls to the incorrect parent in the 60 parental individuals of each species analyzed (deltapar = 0.05 and 0.04 for *X. birchmanni* and *X. malinche*, respectively). The transition probability of the HMM in MSG is determined by the mean genome-wide recombination rate multiplied by a scalar (rfac). We set the recombination rate to 240 based on the estimate of approximately 1 recombination event per chromosome per meiosis (24, 1 cM/378 kb, *Walter et al., 2004*), and an a prior expectation of at least 10 generations of hybridization. We then increased the recombination factor step-wise to the maximum value that did not induce false breakpoints in parental individuals; we determined that rfac could be set to 3 without leading to spurious ancestry calls for parental individuals.

At some point, ancestry blocks will be too small for the HMM to detect given our density of ancestry informative markers. To determine the ancestry block size at which sensitivity decreases, we used the pseudogenomes to generate 25 Mb chromosomes with a homozygous block for the alternate parent randomly inserted (40, 60, 80, 100, 120 kb). We simulated 100 replicates of each size class and generated 1 million reads at MseI sites genome-wide. We then analyzed these simulated individuals using the MSG pipeline and determined whether the homozygous segment was detected. Based on our simulations, we determined that we had low power to detect ancestry blocks smaller than 80 kb (probability of detection ≤80%) and high power to detect blocks 120 kb or larger (probability of detection ≥97%). To determine how much of the genome we are failing to detect in small ancestry segments, we fit an exponential distribution to the observed block sizes in the real data for each parent and for each population, generated samples from an exponential distribution with the lambda of the observed distribution, and determined what percent of bases pairs were found in ancestry blocks below our detection threshold. Based on this analysis, we determined that in both hybrid zones, less than 5% of base pairs in the genome are likely to fall into undetectable segments.

As an additional evaluation of MSG's effectiveness in genotyping hybrid genomes with similar properties to ours, we simulated a 25 Mb admixed chromosome for 100 individuals, drawing ancestry size blocks from the block size distribution observed in the real data. We then generated MSG data in silico for each simulated individual (1 million reads), and compared MSG ancestry calls to true ancestry. We found that on average 91.4% of raw calls were made to the correct genotype; if ambiguous calls were excluded (posterior probability ≤0.95, 7% of sites), MSG's accuracy increased to >98%. The median size of regions for which incorrect calls were made was 29 kb, much smaller than the median block size in the whole data set.

## Acknowledgements

We would like to thank Molly Przeworski, Graham Coop, and Priya Moorjani for statistical advice, Bridgett vonHoldt, Stephen Wright, and Ying Zhen for reading the manuscript, Dale Nyholt for providing the source code for matSpDlite, John Postlethwait for access to the *Xiphophorus maculatus* genome, and the federal government of Mexico for permission to collect fish.

## Additional information

### Funding

| Funder | Grant reference number | Author |
| --- | --- | --- |
| National Science Foundation | IOS-0923825 | Gil G Rosenthal |
| National Science Foundation | GRFP Grant No. DGE0646086 | Molly Schumer |

The funders had no role in study design, data collection and interpretation, or the decision to submit the work for publication.

## Author contributions

MS, Conception and design, Acquisition of data, Analysis and interpretation of data, Drafting or revising the article; RC, GGR, PA, Conception and design, Analysis and interpretation of data, Drafting or revising the article; DLP, RD, Acquisition of data, Analysis and interpretation of data, Drafting or revising the article

## Ethics

Animal experimentation: The procedures used in this study were approved by the Institutional Animal Care and Use Committee at Texas A&M University (Protocols # 2010-111 and 2012-164). Procedures were designed to minimize animal stress and suffering by using anesthesia during fin clipping and minimal handling of the fish. Samples were collected from the wild under Mexican federal collector's license FAUT-217 to W Scott Monks.

# Additional files

## Supplementary file

• Supplementary file 1. (A) Pairs of regions in significant linkage disequilibrium (FDR 5%). (B) Pairs of LD regions (FDR 5%) that have single-gene resolution. (C) Divergence analysis for full data set including pairs heterospecific in one or both populations.

## Major datasets

The following datasets were generated:

| Author(s) | Year | Dataset title | Dataset ID and/or URL | Database, license, and accessibility information |
|---|---|---|---|---|
| Schumer M, Cui R, Powell D, Dresner R, Rosenthal GGR, Andolfatto P | 2014 | Reduced representation data for *Xiphophorus* malinche-birchmanni hybrids | http://www.ncbi.nlm.nih.gov/sra/?term=SRX544941 | Publicly available at NCBI Sequence Read Archive. |
| Schumer M, Cui R, Powell D, Dresner R, Rosenthal GGR, Andolfatto P | 2014 | High-resolution Mapping Reveals Hundreds of Genetic Incompatibilities in Hybridizing Fish Species | doi: 10.5061/dryad.q6qn0 | Available at Dryad Digital Repository under a CC0 Public Domain Dedication. |

The following previously published datasets were used:

| Author(s) | Year | Dataset title | Dataset ID and/or URL | Database, license, and accessibility information |
|---|---|---|---|---|
| Schumer M, Cui R, Boussau B, Walter W, Rosenthal G, Andolfatto P | 2012 | Genome sequencing of *Xiphophorus* birchmanni | http://www.ncbi.nlm.nih.gov/sra/?term=SRX201248 | Publicly available at NCBI Sequence Read Archive. |
| Cui R, Schumer M, Kruesi K, Walter R, Andolfatto P, Rosenthal G | 2013 | *Xiphophorus* malinche whole genome sequences | http://www.ncbi.nlm.nih.gov/sra/?term=SRX246515 | Publicly available at NCBI Sequence Read Archive. |

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
