## [Decision Letter]

Thank you for sending your work entitled “High-resolution Mapping Reveals Hundreds of Genetic Incompatibilities in Hybridizing Fish Species” for consideration at *eLife*. Your article has been favorably evaluated by a Senior editor (Detlef Weigel) and 3 reviewers, one of whom, Gil McVean, is a member of our Board of Reviewing Editors.

The Reviewing editor and the other reviewers discussed their comments before we reached this decision, and the Reviewing editor has assembled the following comments to help you prepare a revised submission.

The genetic architecture of species isolation is a fascinating area and, as a result, there are still many things we don't know about the process. This paper exploits a fantastic natural experiment – multiple independent origins of hybrids between two species of fish that live in different thermal zones of Mexican rivers – and uses genome-wide genotyping to look at patterns of inter- and intra-specific LD within the hybrids. You conclude that 100s of loci are important in causing reproductive isolation (or at least are involved in epistatic fitness interactions), though none are identified directly and the evidence is circumstantial.

These results are potentially very interesting, because they point to many, relatively weak interactions affecting fitness arising from hybridisation. However, in its current form, there are a number of key limitations that need to be addressed.

1) Confirmation of LD not being driven by physical linkage. Long-range LD could be an artefact resulting from mis-assembly or mis-mapping – small-scale transpositions and duplications, or simply because the Xiphophorus maculates genome as a typical next gen draft might have erroneous linkages. You need to show that the regions you identify as being in long range LD are also linked to their supposedly local neighbors.

2) Accuracy of dating hybridisation. If you look at the hybrid genomes in Figure 1, there are what look like big blocks of ancestry from the non-dominant species, but these seem to be interspersed with brief switches back and forth between the parental types. The reported length of tracts is quite small, implying an old hybridisation, but there is a worry that this is due to over-switching in the ancestry painting. We would like to see an analysis more like that of the Hellenthal et al. recent Science paper in humans, where the decay of parental ancestry along the genome is used, rather than the length of blocks inferred directly. Genotyping error, gene conversion, mis-assembly and segregating deletions might all confuse the simple painting approach used and, because so many of the inferences are critically dependent on the age of the hybridisation event, it is important to be robust to any technical problems. We'd also like to see some evidence that a single hybridisation event fits the data well.

3) Predictions of the BDMI model. A key prediction of the 2-locus BDM model is that hybrid incompatibilities will be asymmetric. In the case where the ancestor is AABB, species 1 is AAbb, and species 2 is aaBB, only one of the heterospecific homozygous combinations suffers (aabb) – because the other is the ancestral genotype. For loci with significant LD, you find a deficit of heterospecific, homozygous genotypes, which might be seen even if only one of the two genotypes (aabb) is selected against. However, it is unclear to us if the selection model you propose to explain the observed deviations from expected genotype frequencies is consistent with what is predicted under the BDM model. In particular, it is difficult to understand why there might be selection against all single heterozygote combinations, but not against the double heterozygote (AaBb). And why do you group “opposite homozygotes” together? Again, the BDM model says that only one of these should be underrepresented: aabb. In fact, depending on the predominant direction of interspecific gene flow, AABB might even be favored (e.g., in the scenario above, if most backcrosses are to species 1). Given that gene flow is highly asymmetric in both populations, we wonder if this might explain the prevalence of negative LD associated loci. In general, this asymmetric gene flow warrants further explanation because it will affect the frequencies of genotypes at putatively incompatible loci.

4) Confirmation that a neutral model cannot explain the data. There is a literature on the distribution of ancestry block sizes and the effects of drift and selection on expectations, but it does not appear here (many of these papers trace back to Fisher's junction theory). Given this literature and the simulation results reported here that selection of near zero explains the data well, we wonder whether or not you can exclude the possibility of no selection acting. Can the data also be explained by the neutral fixation of ancestry blocks in a smallish, drifty hybrid population following admixture? The description of the simulations of selection makes no mention of drift, but this is likely a potent force in an admixed population.

5) Representation of models of speciation and prior work. Throughout the manuscript, you neither clearly define nor distinguish between intrinsic and extrinsic postzygotic reproductive isolation (e.g., Abstract). In its classic form, the BDM model specifically describes how intrinsic postzygotic isolation evolves. Extrinsic postzygotic isolation might also evolve via this scenario (i.e., an environmentally-dependent BDMI), or instead by strictly additive gene action. Adding to the confusion, you conflate BDMIs with prezygotic factors. To improve clarity, you might want to describe the BDMI model in its traditional form, and then use a more general term like “genetic incompatibilities” or “isolation loci” to describe heterospecific interactions that prevent zygote formation (e.g., egg-sperm incompatibilities) or lead to extrinsic isolation. We certainly agree that all of these types of genetic incompatibilities might be important for reproductive isolation between these species. Perhaps this should be the starting point, then: intrinsic hybrid sterility/inviability is weak in these species, so what maintains them in the face of considerable gene flow? What other mechanisms might generate the observed patterns of LD and genotype combinations? Similarly, while BDMI are a prominent model for the origin of incompatibilities, theory suggests they do not lead to effective isolating barriers (e.g., [22], Evolution 4: 1027-1035; [23] doi: 10.1098/rstb.2011.0196). This does not invalidate the analysis of LD in hybrids, but it does provide prior expectations that even strong BDMI selection is not expected to leave strong effects on hybrid populations. It seems like this expectation is worth incorporating in the manuscript. One other concern is that you somewhat overstate the novelty of your study. Contrary to the claims, there is quite a bit known about the number and distribution of BDMIs between populations and species. In addition to a series of *Drosophila* studies that use deficiency mapping and/or introgression lines to examine genome-wide BDMIs (including the Presgraves and Masly & Presgraves papers they cite), several other studies have characterized genome-wide patterns of transmission ratio distortion or LD. You cite sunflower and mouse studies, but also see [25] (rice), [42] (tomato), [64] (*C. briggsae*), Gagnaire et al. 2012 (whitefish), [8] (*D. melanogaster*).

---

## [Author Response]

*1) Confirmation of LD not being driven by physical linkage. Long-range LD could be an artefact resulting from mis-assembly or mis-mapping* – *small-scale transpositions and duplications, or simply because the Xiphophorus maculates genome as a typical next gen draft might have erroneous linkages. You need to show that the regions you identify as being in long range LD are also linked to their supposedly local neighbors*.

We demonstrate that regions in cross-chromosomal LD are in stronger LD with neighboring markers on the same chromosome (outside of the region in cross-chromosomal LD). We include details on this analysis in the Materials and methods section (and see Figure 4—figure supplement 4).

The reviewers raise the concern that small-scale duplications (either true duplications or due to mis-assemblies) could generate false positives. We think this is unlikely because in the MSG pipeline, reads that cannot be uniquely mapped are excluded. This suggests that the major problem would stem from duplications in which only one copy is represented in the genome assembly. However, since reads from both copies would map to the single copy represented in the genome assembly, we expect that genotypes at this region would not strongly correlate with genome regions that are physically linked to the missing duplicate. We also expect that we would observe abnormal patterns of local LD decay in such regions, but this is generally not observed (see Figure 4—figure supplement 4).

*2) Accuracy of dating hybridisation. If you look at the hybrid genomes in*
Figure 1*, there are what look like big blocks of ancestry from the non-dominant species, but these seem to be interspersed with brief switches back and forth between the parental types. The reported length of tracts is quite small, implying an old hybridisation, but there is a worry that this is due to over-switching in the ancestry painting. We would like to see an analysis more like that of the Hellenthal et al. recent Science paper in humans, where the decay of parental ancestry along the genome is used, rather than the length of blocks inferred directly. Genotyping error, gene conversion, mis-assembly and segregating deletions might all confuse the simple painting approach used and, because so many of the inferences are critically dependent on the age of the hybridisation event, it is important to be robust to any technical problems. We'd also like to see some evidence that a single hybridisation event fits the data well*.

We agree that this is a problem and we now use the method described in [27] to estimate hybrid zone age. Interestingly, these estimates better agree with estimates based on historical records (assuming two generations per year). In particular, our estimate of hybrid zone age is now consistent with reports that no hybrids were found in 1988 (Rauchenberger et al. 1990) but were abundant by 1997 (63). We cannot exclude the possibility that multiple migration events have occurred, but investigate the potential impact of assuming a single migration event by simulation. Because there are many possible migration scenarios, we focus our analysis on demonstrating that even high levels of continuous migration will not strongly inflate our false discovery rate.

*3) Predictions of the BDMI model. A key prediction of the 2-locus BDM model is that hybrid incompatibilities will be asymmetric. In the case where the ancestor is AABB, species 1 is AAbb, and species 2 is aaBB, only one of the heterospecific homozygous combinations suffers (aabb)* – *because the other is the ancestral genotype. For loci with significant LD, you find a deficit of heterospecific, homozygous genotypes, which might be seen even if only one of the two genotypes (aabb) is selected against. However, it is unclear to us if the selection model you propose to explain the observed deviations from expected genotype frequencies is consistent with what is predicted under the BDM model. In particular, it is difficult to understand why there might be selection against all single heterozygote combinations, but not against the double heterozygote (AaBb). And why do you group “opposite homozygotes” together? Again, the BDM model says that only one of these should be underrepresented: aabb. In fact, depending on the predominant direction of interspecific gene flow, AABB might even be favored (e.g., in the scenario above, if most backcrosses are to species 1)*.

We thank the reviewers for pointing out incorrect assumptions made in our previous simulations of BDMIs, which modeled the average effects of possible BDMI fitness matrices instead of drawing loci randomly from each of them. In addition, we now make no assumptions about the relative fitness of intermediate genotypes except that genotype combinations that did not exist in the mutational path of either species are under equal selection. We also include simulations and discussion of a non-BDMI model of hybrid incompatibility (for example a co-evolutionary model, see Figure 5—figure supplement 1); our results suggest that this model makes very similar predictions.

*Given that gene flow is highly asymmetric in both populations, we wonder if this might explain the prevalence of negative LD associated loci*.

The reviewers bring up the point that under a BDMI model one heterospecific-homozygous genotype combination (MM_BB or BB_MM) will not be under negative selection, and suggest that this could be the cause of heterospecific associations in Tlatemaco, especially with skewed ancestry proportions. Though this can cause differences in ancestry between sites in simulations, we do not see significantly negative R between genotypes in simulations of this process regardless of initial admixture proportion. We interpret this as the result of conflicting signal from intermediate genotypes (for e.g., BM_BB or MM_BM) that have positive ancestry correlations.

*In general, this asymmetric gene flow warrants further explanation because it will affect the frequencies of genotypes at putatively incompatible loci*.

By “asymmetric gene flow” we believe the reviewers are referring to the fact that hybrid populations are skewed in their genome wide ancestry and that this requires some explanation. A simple explanation of this pattern is that the populations did not admix at 50-50 admixture proportions. A more complex possible explanation is that selection could drive this skew through many negative epistatic interactions in the genome (alluded to in [23]). We now include this in brief in the Discussion section. Regardless of the cause, we believe we have accounted for the effects of this skew in our analysis; permutation of the data preserves the ancestry skew at each site and in coalescent simulations of hybrid populations we explicitly modeled skewed admixture proportions to match the data.

*4) Confirmation that a neutral model cannot explain the data. There is a literature on the distribution of ancestry block sizes and the effects of drift and selection on expectations, but it does not appear here (many of these papers trace back to Fisher's junction theory). Given this literature and the simulation results reported here that selection of near zero explains the data well, we wonder whether or not you can exclude the possibility of no selection acting. Can the data also be explained by the neutral fixation of ancestry blocks in a smallish, drifty hybrid population following admixture? The description of the simulations of selection makes no mention of drift, but this is likely a potent force in an admixed population*.

We now include hybrid population size as a parameter in our approximate Bayesian simulations to incorporate joint effects of drift and selection (Figure 5), and include posterior predictive simulations based on the joint posterior distribution of all parameters to check the consistency of our estimates with the data. The 95% CI interval of our estimate of s does not include s=0 (see Figure 5). We find that the selection coefficient and hybrid population size are correlated in our posterior distributions, but that stronger s is required in small populations to generate the observed two-locus genotype frequencies (see Figure 5—figure supplement 1). In addition, we perform coalescent simulations using the MAP estimate of hybrid population size from the posterior distribution to investigate how hybrid population size might affect our false discovery rate and we find that our expected false discovery rate under this neutral demographic scenario is approximately 10%.

*5) Representation of models of speciation and prior work. Throughout the manuscript, you neither clearly define nor distinguish between intrinsic and extrinsic postzygotic reproductive isolation (e.g., Abstract). In its classic form, the BDM model specifically describes how intrinsic postzygotic isolation evolves. Extrinsic postzygotic isolation might also evolve via this scenario (i.e., an environmentally-dependent BDMI), or instead by strictly additive gene action. Adding to the confusion, you conflate BDMIs with prezygotic factors. To improve clarity, you might want to describe the BDMI model in its traditional form, and then use a more general term like “genetic incompatibilities” or “isolation loci” to describe heterospecific interactions that prevent zygote formation (e.g., egg-sperm incompatibilities) or lead to extrinsic isolation. We certainly agree that all of these types of genetic incompatibilities might be important for reproductive isolation between these species*.

We now clearly present different models of hybrid incompatibility in the text and Figure 5—figure supplement 1, distinguishing between the BDMI model and other possible models of selection against hybrids.

*Perhaps this should be the starting point, then: intrinsic hybrid sterility/inviability is weak in these species, so what maintains them in the face of considerable gene flow? What other mechanisms might generate the observed patterns of LD and genotype combinations? Similarly, while BDMI are a prominent model for the origin of incompatibilities, theory suggests they do not lead to effective isolating barriers (e.g.,*
[22]*, Evolution 4: 1027-1035;*
[23] doi: 10.1098/rstb.2011.0196*). This does not invalidate the analysis of LD in hybrids, but it does provide prior expectations that even strong BDMI selection is not expected to leave strong effects on hybrid populations. It seems like this expectation is worth incorporating in the manuscript*.

We have added discussion on theoretical work that suggests that BDMIs do not lead to effective isolating barriers, especially in the context of our findings that suggest reduced gene flow at the regions we detect.

*One other concern is that you somewhat overstate the novelty of your study. Contrary to the claims, there is quite a bit known about the number and distribution of BDMIs between populations and species. In addition to a series of* Drosophila *studies that use deficiency mapping and/or introgression lines to examine genome-wide BDMIs (including the Presgraves and Masly & Presgraves papers they cite), several other studies have characterized genome-wide patterns of transmission ratio distortion or LD. You cite sunflower and mouse studies, but also see*
[25]
*(rice),*
[42]
*(tomato),*
[64]
*(*C. briggsae*), Gagnaire et al. 2012 (whitefish),*
[8]
*(*D. melanogaster*)*.

We now more clearly place our study in the context of past work and add discussion of the suggested papers.